# Autoencoder and Partially Impossible Reconstruction Losses

**DOI:** 10.3390/s22134862

**Published:** 2022-06-27

**Authors:** Steve Dias Da Cruz, Bertram Taetz, Thomas Stifter, Didier Stricker

**Affiliations:** 1Basics & Mathematical Models, IEE S.A., L-7795 Bissen, Luxembourg; thomas.stifter@iee.lu; 2Department of Computer Science, University of Kaiserslautern, D-67653 Kaiserslautern, Germany; didier.stricker@dfki.de; 3Augmented Vision, German Research Center for Artificial Intelligence (DFKI), D-67663 Kaiserslautern, Germany; bertram.taetz@dfki.de

**Keywords:** autoencoder, generalization, Sim2Real, illumination, reconstruction, sampling, uncertainty

## Abstract

The generally unsupervised nature of autoencoder models implies that the main training metric is formulated as the error between input images and their corresponding reconstructions. Different reconstruction loss variations and latent space regularizations have been shown to improve model performances depending on the tasks to solve and to induce new desirable properties such as disentanglement. Nevertheless, measuring the success in, or enforcing properties by, the input pixel space is a challenging endeavour. In this work, we want to make use of the available data more efficiently and provide design choices to be considered in the recording or generation of future datasets to implicitly induce desirable properties during training. To this end, we propose a new sampling technique which matches semantically important parts of the image while randomizing the other parts, leading to salient feature extraction and a neglection of unimportant details. The proposed method can be combined with any existing reconstruction loss and the performance gain is superior to the triplet loss. We analyse the resulting properties on various datasets and show improvements on several computer vision tasks: illumination and unwanted features can be normalized or smoothed out and shadows are removed such that classification or other tasks work more reliably; a better invariances with respect to unwanted features is induced; the generalization capacities from synthetic to real images is improved, such that more of the semantics are preserved; uncertainty estimation is superior to Monte Carlo Dropout and an ensemble of models, particularly for datasets of higher visual complexity. Finally, classification accuracy by means of simple linear classifiers in the latent space is improved compared to the triplet loss. For each task, the improvements are highlighted on several datasets commonly used by the research community, as well as in automotive applications.

## 1. Introduction

Autoencoder (AE) and closely related encoder–decoder models have seen numerous successful applications, partially due to their bottleneck design, but also because of resulting advantageous properties, e.g., disentanglement [1], feature reduction and compression. The aforementioned desirable characteristics can either be achieved using dedicated priors in the latent space, e.g., a Gaussian prior [2], triplet loss [3], discretization [4], or by non-identical input-target pairs, e.g., in case of denoising and inpainting [5] or super-resolution [6]. It is also common to pretrain autoencoder models on large datasets of general images to learn a meaningful and generic feature extraction such that the resulting encoder can be fixed during a second fine-tuning stage [7]. During the latter, an additional classifier head using the extracted features as input can be trained on the dataset of the actual task to be solved [8].

The new training methods presented in this work combine ideas of most of the aforementioned approaches. The different variations of the newly introduced partially impossible reconstruction loss (PIRL) are novel sampling strategies whose goals are to match semantically important and similar parts between target and input images and randomize the other parts. The latter can be as simple as selecting a different image of the same class as the input image. In a more advanced step, one can select a new variation of the same scene, e.g., an identical scene under different illumination conditions, as the target image. The features which can be considered as important and unimportant depend on the task to be solved: in case of image classification, the discriminative characteristics of the different classes are important, while we might not be interested in facial landmarks, human poses, illumination or backgrounds such that these features can be considered unimportant.

In this work, we investigate and highlight in detail the benefits and downsides of the PIRL loss. We first use simpler datasets commonly used by the research community to show that the PIRL loss produces a good latent space representation on par with using the triplet loss. We will further show that the PIRL loss can be used to perform image illumination and scene normalization. Notwithstanding this advantage, a combination of the PIRL and triplet loss can provide additional advantages in case of nearest neighbour retrieval. For the case of synthetic to real generalization, we combine the PIRL loss with a feature extractor to improve generalization and robustness. Lastly, we show that the PIRL loss can also improve uncertainty and out-of-distribution (OOD) estimations across many different datasets, particularly for datasets of higher visual complexity. For each investigation, we showcase the previously mentioned design choices and their advantages in an automotive application to highlight the applicability of the provided insights. The latter considers rear seat occupant classification in the vehicle interior, i.e., classifying whether adults, child seats, infant seats or objects are occupying each individual seat position. Each experiment is combined with an ablation study to highlight the effect of the different design choices and to compare the performance against commonly used CNN classification models being contentpre-trained and fine-tuned. Our proposed sampling strategy can easily be used with existing autoencoder models, reconstruction losses and sampling strategies. The code implementation to reproduce all experiments is made publicly available as well—see Appendix A.

## 2. Related Works

Sampling: The most used sampling strategy is probably the triplet loss [9] which implicitly uses the label information in the latent space by retrieving positive and negative samples (with regard to the target label of the input images) and forcing images of the sample label to be clustered together and images of different labels to be pushed away. Other latent space losses are the angular loss [10], contrastive loss [11] and n-pair loss [12]. We will compare our method against the triplet loss, since it is the most commonly used. Notice that our sampling strategy can be combined with any of the aforementioned losses such that it can be used isolated or as an extension.

Consistency in latent space: Existing encoder–decoder based methods try to represent the information from multiple domains [13] or real–synthetic image pairs [14] identically in the latent space, e.g., by enforcing the latent vectors to be close together. However, these approaches force networks to reconstruct some (or all) of the images correctly in the decoder part. Forcing an encoder–decoder to represent two images (e.g., the same scenery, but different lightning) identically in the latent space, yet simultaneously forcing it to reconstruct both input images correctly, implies an impossibility: The decoder cannot reconstruct two different images using the same latent space. Antelmi et al. [13] adopted a different encoder–decoder for each domain, but as illumination changes are continuous and not discrete, we cannot have a separate encoder or decoder for each illumination.

It is believed that scene decomposition into meaningful components can improve the transfer performance on a wide range of tasks [15]. Although datasets such as CLEVR [16] and Objects Room [15] exist, they are limited to toy examples and lack increased visual complexity. Instead, we consider the vehicle interior, which is still tractable due to the limited background variation and visually more complex due to human models.

Shadow removal and relighting: Recent advances in portrait shadow manipulation [17] try to remove shadows cast by external objects and to soften shadows cast by the facial features. While the proposed method can generalize to images taken in the wild, it has problems for detailed shadows and it assumes that shadows either belong to foreign or facial features. Most importantly, it assumes facial images as input and exploits the detection of facial landmarks and their symmetries to remove the shadows. Other shadow removal methods [18,19] are limited to simpler images. The backgrounds and illumination are usually uniform and they contain a single connected shadow. Moreover, the availability of shadow and shadow-free image pairs provides the means of a well defined ground truth. This is is not possible for more complex scenes and illumination conditions for which a ground truth is not available or is impossible to define. Image relighting [20,21] could potentially be used to change the illumination of an image to some uniform illumination. However, as noted in [17,20] relighting struggles with foreign or harsh shadows. While it is possible to fit a face to a reference image [22], this option is limited to facial images as well.

Recording identical, or similar, sceneries under different lightning or environmental conditions is a challenging task. Large-scale datasets for identical sceneries under different lightning conditions are currently scarce. The Deep Portrait Relighting Dataset [21] is based on the CelebA-HQ [23] dataset and contains human faces under different illumination conditions. However, the re-illumination has been added synthetically. We instead used the Extended Yale Face Database B [24], which is a dataset of real human faces with real illumination changes. While cross-seasons correspondence datasets prepared according to Larsson et al. [25] and based on RobotCar [26] and CMU Visual Localization [27] could be used for our investigation, the correspondences are usually not exact enough to have an identical scene under different conditions. Moreover, dominantly changing vehicles on the street induce a large difference in the images. These datasets stem from the image correspondence search and SLAM community. We adopt the Webcam Clip Art [28] to include a dataset for the exterior environment with changing seasons and day times. The latter contains webcam images of outdoor regions from different places all over the world.

Synthetic to real: There have been successful applications of reinforcement learning systems being trained in a simulated environment and deployed to a real one, for example, by combining real and synthetic data during training [29,30]. However, these approaches can take into account temporal information and action–reaction causalities, while in this work we use independent frames only. A good overview on reinforcement-learning-based simulation to real transferability is provided in Zhao et al. [31]. Another line of research uses generative adversarial networks (GAN) to make synthetic images look like real images or vice versa [32,33]. This requires both synthetic and real images, whereas we focus on training on synthetic images only. Part of our methodology is related to domain randomization [34], where the environment is being randomized, but the authors deployed this to object detection and the resulting model needs to be fine-tuned on real data. A similar idea of freezing the layers of a pretrained model was investigated for object detection [35], but neither with a dedicated sampling strategy nor in the context of autoencoders. Another work focuses on localization and training on synthetic images only [36], though the applicability is only tested on simple geometries. Recent advances on synthetic to real image segmentation [37,38] on the VisDA [39] dataset show a promising direction to overcome the synthetic to real gap; however, this cannot straightforwardly be compared against the investigation in this work, since we are focusing on autoencoders and their generative nature. Others rely on the use of real images during training for reducing the synthetic to real gap for autoencoders [14,40].

Uncertainty estimation: A lot of research [41] is focused on estimating the uncertainty of a model’s prediction regarding OOD or anomaly detection. However, only a few studies consider the use of autoencoders for assessing uncertainty: autoencoders can be combined with normalizing flow [42], refactor ideas from compressed sensing [43] or use properties of Variational Autoencoders [44,45]. More commonly, autoencoders are used for non image based datasets [46,47]. Other deep learning approaches are based on evidential learning [48], Bayesian methods [49], Variational Bayes [50] or on Hamiltonian Monte Carlo [51]. Additionally, non-deep-learning approaches have shown significant success, but are less scalable. Some examples are Gaussian Processes [52] or approaches based on support vector machines [53]. Since our approach borrows ideas from MC dropout [54], we limit our comparison against the latter and the commonly used deep learning golden standard of an ensemble of trained models [55].

Vehicle interior datasets: Some previous works have been investigating occupant classification [56,57], seat-belt detection [58] or skeletal tracking [59] in the passenger compartment, but, to the best of our knowledge, no dataset was made publicly available. Publicly available realistic datasets for the vehicle interior are scarce. Some exceptions are the recently released AutoPOSE [60] and DriveAHead [61] datasets for driver head orientation and position, Drive&Act [62] a multi modal benchmark for action recognition in automated vehicles and TICaM [63] for activity recognition and person detection. However, these datasets provide images for a single vehicle interior only.

In this work, we use SVIRO [64] and its extensions SVIRO-Illumination [65], SVIRO-NoCar [66] and SVIRO-Uncertainty [67]. Although SVIRO is a synthetic dataset, it was designed specifically to test the transferability between different vehicles across multiple tasks. Moreover, together with all its extensions, including different settings and splits, it provides a common framework to investigate all the aforementioned topics. The applicability to real infrared [64,68] and depth images [69] was shown. Recent studies showed the importance of synthetic data for the automotive industry [34,70]. We use TICaM for real images for the vehicle interior.

## 3. Methods

We provide an overview of all the methods used in this work. We start by formulating a framework and common language and will then explain the different building blocks which are presented as standalone, but which can also be combined.

### 3.1. Problem Statement

Consider $N_{s}$ sceneries and $N_{v}$ variations of the same scenery, e.g., the same scenery under different illuminations, with different backgrounds or under different data augmentation transformations. Most commonly used datasets will have $N_{v}\;=\;1$ unless they are cleverly augmented by transformations (i.e., preserving the important semantics) to adopt our proposed sampling strategy. Let $X\;=\;\{X_{i}^{j}\;|\;1\leq i\leq N_{v},\;1\leq j\leq N_{s}\}$ denote the training data, where each $X_{i}^{j}\in R^{C\times H\times W}$ is the *i*th variation of scene *j* consisting of *C* channels and with a height of *H* and width of *W*. Let $X^{j}\;=\;\left\{X_{i}^{j}\;|\;1\leq i\leq N_{v}\right\}$ be the set of all variations *i* of scenery *j* and $Y\;=\;\{Y^{j}\;|\;1\leq j\leq N_{s}\}$ be the corresponding target classes of the scenes of X. Notice that the classes remain constant for the variations *i* of each scene *j*. This is important for classification tasks and it needs to be adapted in case our sampling strategy is adopted for other tasks, i.e., landmark detection.

Let $e_{\phi }$ be the encoder and $d_{\theta }$ be the decoder. We define $d_{\theta }\left(e_{\phi }\left(\cdot \right)\right)\;=\;{\mathrm{de}}_{\theta ,\phi }\left(\cdot \right)$. An autoencoder using the vanilla reconstruction loss can be formulated for a single input sample as
(1)$$L_{R}\left(X_{i}^{j};\;\theta ,\;\phi \right)\;=\;r\left(d_{\theta }\left(e_{\phi }\left(X_{i}^{j}\right)\right),\;X_{i}^{j}\right)\;=\;r\left({\mathrm{de}}_{\theta ,\;\phi }\left(X_{i}^{j}\right),\;X_{i}^{j}\right)$$
where r(·, ·) computes the error between input and reconstruction. We use the structural similarity index measure (SSIM) [71], but our method is not limited to the latter, e.g., one could also use mean squared error (MSE) or perceptual loss [72].

### 3.2. Sampling Strategy: Partial Impossible

The main contribution of this work is the introduction of a novel sampling strategy, for which we provide two variations. The first one is the partially impossible reconstruction loss (PIRL) as initially introduced by Dias Da Cruz et al. [65] for illumination normalization. For this variation of the PIRL loss, we select as target for each input image a variation of the latter: we can select the same scene under different illumination and/or different backgrounds. However, the method is not limited to the latter—one could, for example, also simply augment input and target images differently. It is important that the features we want to become invariant against are being varied and the features we assess as discriminative are being kept identical or similar. This also means that the target of the image should not be changed (e.g., if flipping the images would change the label). In this work, for sampling the individual elements of a batch, we randomly select two images for each scene, one as input and the other one as the target. This sampling strategy preserves the semantics while varying the unimportant features such that the model needs to focus on what remains constant. This leads to the formulation:(2)$$L_{R,I}\left(X_{a}^{j};\;\theta ,\;\phi \right)\;=\;r\left({\mathrm{de}}_{\theta ,\;\phi }\left(X_{a}^{j}\right),\;X_{b}^{j}\right),$$
for random $a,b\in [0,\;N_{v}]$ and a≠b. We refer to models using this variation of the PIRL by prepending an *I*, i.e., we call it I-PIRL and the models using it, for example, I-AE.

### 3.3. Sampling Strategy: Partial Impossible Class Instance

The previously introduced PIRL will lead to good results, particularly for normalization; however, it can be challenging to apply it to a lot of commonly recorded datasets. Many datasets do not provide multiple variations for each piece of scenery. Consequently, to adopt the I-PIRL, input and target images need to be transformed manually and differently with respect to the invariances to be learned. However, the latter does not provide as good results as if the dataset yields the required variations. To this end, we introduced a second variation of the previously introduced PIRL which can readily be applied to most existing datasets. Instead of sampling the same scene under a controlled variation (e.g., the same scene under different illumination), we propose to use as a target image a different image of the same class as the input. This approach implicitly uses label information in the input space; however, as we will show, this leads to a better latent space representation. This loss variation causes the model to learn invariances with respect to certain class variations which are not important for the task at hand, e.g., clothes, human poses, textures. This sampling variation is reflected in the reconstruction loss as follows:(3)$$L_{R,II}\left(X_{a}^{j};\;\theta ,\;\phi \right)\;=\;r\left({\mathrm{de}}_{\theta ,\phi }\left(X_{a}^{j}\right),X_{b}^{k}\right),$$
for random $a,b\in [0,\;N_{v}]$, j≠k and $Y^{j}\;=\;Y^{k}$. We refer to this method as partially impossible class instance sampling marked by prepending *II* when it is used, i.e., referring to it as II-PIRL and the models using it as, for example, II-AE. The two sampling variations are visualized in Figure 1.

### 3.4. Structure in the Latent Space: Triplet Loss

Our sampling strategies can easily be combined with other commonly used sampling strategies. In this work, we limit ourselves to the triplet loss to induce regularization and structure in the latent space of autoencoder models [73]. On the one side this can lead to a better nearest neighbour retrieval: the triplet loss uses the label information in the latent space to project images of the same class closely together and images of different classes far away from one another. The distance is usually defined using the $L^{2}$ norm, which causes the latent space to be quasi Euclidean [74]. On the other side, we introduce this formulation to compare our method against it. The triplet loss can be integrated by
(4)$$L_{T}\left(X_{a}^{j};\phi \right)\;=\;\max\left(0,\;{\left\|e_{\phi }\left(X_{a}^{j}\right)\;-\;e_{\phi }\left(X_{b}^{k}\right)\right\|}^{2}\;-\;{\left\|e_{\phi }\left(X_{a}^{j}\right)\;-\;e_{\phi }\left(X_{c}^{l}\right)\right\|}^{2}\;+\;0.2\right),$$
for random $a,\;b,\;c\in [0,\;N_{v}]$, j≠k≠l and $Y^{j}\;=\;Y^{k}\neq Y^{l}$. We refer to this model as triplet autoencoder (TAE) either with or without using the PIRL. In case of the former, we can sample impossible target instances for the positive and negative triplet samples such that the total loss becomes (for some α and β):(5)$$L\left(X_{a}^{j};\;\theta ,\;\phi \right)\;=\;\alpha L_{T}\left(X_{a}^{j};\phi \right)\;+\;\beta \left(L_{R,II}\left(X_{a}^{j};\;\theta ,\;\phi \right)\;+\;L_{R,II}\left(X_{b}^{k};\;\theta ,\;\phi \right)\;+\;L_{R,II}\left(X_{c}^{l};\;\theta ,\;\phi \right)\right).$$

We chose α = 1 and β = 1 as it provided good results, but a dedicated hyperparameter search can improve the performance even further. An unbalanced optimization can occur depending on which reconstruction loss is used, i.e., in case the triplet loss is much larger or smaller than the sum of the reconstruction losses.

### 3.5. Model Architecture: Extractor Autoencoder

In addition to the proposed variations of the PIRL loss, we will also highlight the benefit of using a feature extractor in case of generalizing from synthetic to real images. This model architecture can be combined with the PIRL loss or be used as a standalone design choice. We propose applying ideas from transfer learning and use a pretrained classification model to extract more general features from the input images. Instead of using the images itself, the extracted features are used as input. Our autoencoder consists of a summarization module which reduces the number of convolutional filters. This is fed to a simple MLP encoder, which is then decoded by a transposed convolutional network. We refer to this model as extractor autoencoder (E-AE). Let $e_{\phi }$ be the encoder, $d_{\theta }$ the decoder and ${\mathrm{ext}}_{\omega }$ be a pretrained classification model, referred to as extractor. For ease of notation, we define $e_{\phi }\left({\mathrm{ext}}_{\omega }\left(\cdot \right)\right)\;=\;{\mathrm{ee}}_{\phi ,\;\omega }\left(\cdot \right)$. The model, using the vanilla reconstruction loss, can be formulated for a single input sample as
(6)$$L_{R}\left(X_{i}^{j};\;\theta ,\;\phi \right)\;=\;r\left(d_{\theta }\left(e_{\phi }\left({\mathrm{ext}}_{\omega }\left(X_{i}^{j}\right)\right)\right),\;X_{i}^{j}\right)\;=\;r\left(d_{\theta }\left({\mathrm{ee}}_{\phi ,\omega }\left(X_{i}^{j}\right)\right),\;X_{i}^{j}\right).$$

This loss can easily be combined with both PIRL loss variations and with the triplet loss.

### 3.6. Model Architecture: Uncertainty Estimation

The use of dropout during training and inferences, called Monte Carlo (MC) dropout, has been introduced [54] to model uncertainty in neural networks without sacrificing complexity or test accuracy for several machine learning tasks. For standard classification or regression models, an individual binary mask is sampled for each layer (except the last layer) for each new training and test sample. Consequently, neurons are dropped randomly such that during inference we sample a function f from a family, or distribution of functions F, i.e., f∈F. Uncertainty and reliability can then be assessed by performing multiple runs for the same input sample *x*, i.e., retrieve ${\left\{f_{j}\left(x\right)\right\}}_{j\in J}$ for J = {1, 2, ⋯,M} for some M≥1. The model’s predictive distribution for an input sample *x* can then be assessed by computing $p\;=\;f\left(x\right)\;=\;\frac{1}{M}\sum_{j\;=\;1}^{M}\mathrm{softmax}\left(f_{j}\left(x\right)\right)$. Uncertainty can be summarized by computing the normalized entropy [75] of the probability vector *p*, i.e., $H\left(p\right)\;=\;-\frac{1}{\log(C)}\sum_{c\;=\;1}^{C}p_{c}\log\left(p_{c}\right)$, where *C* is the number of classes. We use the latter in all our experiments to compute the uncertainty of the prediction and decide based on its value whether a sample is rejected or accepted for prediction or whether the sample is in- or out-of-distribution.

Instead of training the autoencoder model under a standard training regime, we train the model using dropout and enable dropout during inference as well. Hence, we obtain different, but similar, autoencoder models for inference which should behave similarly for training and test samples, but differently and not consistently for novel feature variations in the input space. Let us formulate this intuition more precisely: let *x* be an input sample and F be the family of functions consisting of the autoencoder models learned by using dropout during training and enabling the latter during inference. We repeat inference *M* times, each time sampling a new $f_{j}\in J\;=\;\left\{1,\;2,\;\cdots ,\;M\right\}$. This results in a predictive distribution ${\left\{f_{j}\left(x\right)\right\}}_{j\in J}$. Since we are interested in the variation of the latent space representation, we refrain from using a dropout in the latent space. Finally, this approach can easily be extended using, for example, the II-PIRL loss. We refer to the latter as MC-II-AE and the former as MC-AE.

### 3.7. Model Architecture: Classification

We use the latent space representation to perform a classification for all the previously introduced methods, i.e., we are not interested in the reconstruction and the pixel space. After training the autoencoder model in the first stage, we retrieve the latent space representation for all training samples in the second stage. Then, in the third stage, we train a classifier on the training data latent space representation. For the latter, we use several classifiers in this work: linear classifier, k-nearest neighbour (KNN), support vector machine (SVM), random forest (RForest) and a single hidden layer MLP. Of course, other classifiers and an end-to-end training could also be considered.

## 4. SVIRO

SVIRO [64] is a synthetic dataset for sceneries in the passenger compartment of ten different vehicles. Each vehicle consists of a train (2500 images) and a test (500 images) split: SVIRO uses different objects for the train and test split to evaluate the generalization to unknown vehicle interiors either using known or unknown class instances. SVIRO contains seven different classes: empty seat, occupied and empty infant seat, occupied and empty child seat, adult passenger and everyday object (e.g., bags). Eight vehicles have three seats and two vehicles have two seats (in which case the middle one is treated as empty). We conducted our experiments on the grayscale images (simplified infrared imitations), but depth maps and RGB images, as well as bounding boxes, segmentation masks and keypoints are available as well. Several dataset extensions were created to investigate additional questions in the same framework. Examples for the different datasets are shown in Figure 2.

### 4.1. SVIRO-Illumination

Additional images for three new vehicle interiors were created. For each vehicle, 250 training and 250 test scenes where scenery was rendered under 10 different types of illumination and environmental conditions were randomly generated. There are two versions: one containing only people and a second one including additionally occupied child and infant seats. The models need to generalize to new illumination conditions, humans and textures. There are four possible classes for each seat position (empty, infant seat, child seat and adult) leading to $4^{3}\;=\;64$ classes for the whole image.

### 4.2. SVIRO-NoCar

For learning more general features by using the II-PIRL for synthetic to real generalization, we needed input–target pairs where images are of the same scene, but differ in the properties we want to become invariant to: the dominant background. To this end, we created 5919 synthetic scenes where we placed humans, child and infant seats as if they would be sitting in a vehicle interior, but instead of a vehicle, the background was replaced: each scene was rendered using 10 different backgrounds.

### 4.3. SVIRO-Uncertainty

Finally, to assess model uncertainty and out-of-distribution detection in an industrial application, we extended SVIRO with another dataset. Two additional training datasets for a new vehicle, the Sharan, were created: one using adult passengers only (4384 sceneries and 8 classes—not used in this work) and one using adults, child seats and infant seats (3515 samples and 64 classes). Further, we created fine-grained test sets to assess the reliability on several difficulty levels: (1) only unseen adults, (2) only unseen child and infant seats, (3) unseen adults and unseen child and infant seats, (4) unknown random everyday objects (e.g., dog, plants, bags, washing machine, instruments, tv, skateboard, paintings, …), (5) unseen adults and unknown everyday objects and (6) unseen adults, unseen child and infant seats and unknown everyday objects. In addition to uncertainty estimation within the same vehicle interior, one can use images from unseen vehicle interiors from SVIRO to further test the model’s reliability on the same task, but in novel environments, i.e., vehicle interiors.

## 5. Basic Analysis

We perform an exhaustive investigation in Section 5, Section 6, Section 7 and Section 8 of the different model variations presented in Section 3. The code implementation for all results is made publicly available.

In this section, we start with a basic analysis on datasets commonly used by the research community. This should build an entry point to become familiar with the loss function and to understand the effect on well-known datasets, also in comparison with other methods. We trained vanilla autoencoder (AE), autoencoders with the triplet loss (TAE) and autoencoders with the II-PIRL (II-AE) on MNIST [76], Fashion-MNIST [77], CIFAR10 [78] and GTSRB [79]. We then compare the test accuracy and latent space representation of the different models.

For training on MNIST and Fashion-MNIST we used a latent space of 16, while for all others, we used a latent space of 64. We treat all datasets as grayscale images. In case of RGB images, we used 0.299R + 0.587G + 0.114B, as used by opencv, to transform the RGB images into grayscale ones. Moreover, all images were centre cropped and resized to 64 by 64 pixels. We used a batch size of 64, trained our models for 1000 epochs and did not perform any data augmentation. Classification was performed by means of a linear classifier in the latent space which was trained after the autoencoder model was finished training. The latter gives hints about the quality and separability of the learned latent space representation, and hence feature extraction.

### Results

In Figure 3, we compare the t-SNE projection of the training data latent space representation for the different autoencoder models for different datasets. It can be observed that the II-PIRL produces very clear clusters for the different classes. This is an interesting phenomenon, since the label information is only used implicitly in the pixel space and not explicitly in the latent space as achieved by the triplet loss. In order to assess the representation quantitatively, we consider the test accuracy performance by a linear classifier when trained on the aforementioned latent spaces. The results are reported in Table 1. The linear classifier can exploit the latent space representation at least as well as if the triplet loss is used: the test accuracy is slightly better. Finally, to get an idea about the input–target pairs used for training and the resulting reconstruction of the different methods, we report in Figure 4 reconstructions of the training data after training. It can be observed that the II-AE removes all unnecessary background information (see GTSRB) and learns a mean class representation. The model is not able to learn a meaningful mean class representation for CIFAR10, due to its large intra-class variability; nevertheless, as shown by the latent space representation and quantitative results, the model can still learn a somewhat meaningful separation in the latent space and achieve a better test accuracy as if the triplet loss is used.

## 6. Image and Illumination Normalization

Normalizing images and illumination works well on the training data when the PIRL is used, but generalizing to unseen test images can remain a challenging task if no additional precautions are taken. The illumination is removed from test samples, but the reconstruction of the objects of interest can be less stable. If training data are limited, the encoder–decoder network is mostly used as a compression method instead of a generative model. Consequently, generalizing to unseen variations cannot trivially be achieved. Example of failures are plotted in Figure 5 and Figure 6: it can be observed that the application on test images can cause blurry reconstructions. It turns out that the blurry reconstruction is in fact a blurry version of the reconstruction of its nearest neighbour in the latent space (or a combination of several nearest neighbours). An example of a comparison of the five nearest neighbours for several encoder–decoder models is shown in Figure 7.

Consequently, instead of reconstructing the latent space representation of the encoded test sample, it is more beneficial to reconstruct the latent space representation of the nearest neighbour with respect to the $L^{2}$ norm. However, applying a nearest neighbour search in the latent space of a vanilla autoencoders (AE) or variational autoencoders (VAE) will not provide robust results. This is due to the fact that there is no guarantee that the learned latent space representation follows an $L^{2}$ metric [80]. As the nearest neighbour search is (usually) based on the $L^{2}$ norm, the latter will not always work reliably. To this end, we incorporated a triplet loss, as introduced in Section 3.4, in the latent space of the encoder–decoder model (TAE) instead. This effect is highlighted in Figure 7, where we compare the nearest neighbours of the AE, VAE and TAE.

We centre-cropped the images to the smallest image dimension and then resized it to a size of 224 × 224. We used a batch size of 16, trained our models for 1000 epochs and did not perform any data augmentation. We used a latent space of dimension 16. The model uses the VGG-11 architecture [81] for the encoder and reverses the layers together with nearest neighbour up-sampling for the decoder.

### 6.1. Extended Yale Face Database B

The Extended Yale Face Database B [24] contains images of 28 human subjects under nine poses. For each pose and human subject, the same image is recorded under 64 illumination conditions. We considered the full-size image version and used 25 human subjects for training and 3 for testing. We removed some of the extreme dark (no face visible) illumination conditions. Example images from the dataset are plotted in Figure 5. For the triplet sampling, we chose as a positive sample an image with the same head pose, and for the negative sample, an image with a different head pose. We report qualitative results in Figure 5. The model is able to remove lightning and shadows from the training images, but the vanilla reconstruction on test samples can be blurry. We are not using the centre-cropped images, which makes the task more complicated, because the head is not at the same position for each human subject. Nevertheless, the model is able to provide a nearest neighbour with a similar head pose and position.

### 6.2. Webcam Clip Art

The Webcam Clip Art [28] dataset consists of images from 54 webcams from places all over the world. The images are recorded continuously, such that the same scenery is available for different day times, seasons and weather conditions. We randomly selected 100 sceneries for each region. Example images are provided in Figure 8. For the triplet sampling, we chose as positive sample an image from the same location, and for the negative sample, an image from a different location. Each landscape and building arrangement undergoes unique shadow, illumination and reflection properties. The generalization to unknown places under unknown illumination conditions is thus too demanding to be deduced from a single input image. Hence, we report results on training samples only in Figure 8. The model removes the illumination variations and shadows from the images. Moreover, rivers, oceans and skies, as well as beaches, are smoothed out. Most of the people and cars are removed and replaced by the background of the scenery. The features of the salient objects are preserved when their position remains constant in each image, e.g., see Figure 8 for vehicles being removed if not contained in each image.

### 6.3. SVIRO-Illumination

For the triplet loss sampling, we chose the positive sample to be of the same class as the anchor image (but from different scenery) and the negative sample to differ only on one seat (i.e., change only the class on a single seat with regard to the anchor image). Images of three empty seats do not contain any information which could mislead the network, so to make it more challenging, we did not use them as negative samples. After training, the encoder–decoder model learned to remove all the illumination and environmental information from the training images. See Figure 6 for an example of how images from the same scenery, but under different illumination, are transformed. Sometimes, test samples are not reconstructed reliably. However, due to the triplet loss and nearest neighbour search, we can preserve the correct classes and reconstruct a clean image: see Figure 6 for an example. The reconstruction of the test image latent vector produces a blurry person, which is usually a combination of several nearest neighbours. We want to emphasize that the model is not learning to focus the reconstruction to a single training image for each piece of scenery. In Figure 9 we searched for the closest and furthest (with regard to SSIM) input images of the selected scenery with regard to the reconstruction of the first input image. Moreover, we selected the reconstruction of all input images which is furthest away from the first one to get an idea about the variability of the reconstructions inside a single piece of scenery. While the reconstructions are stable for all images of a piece of scenery, it can be observed that the reconstructions are far from all training images. Hence, the model did not learn to focus the reconstruction to a single training sample, but instead learned to remove all the unimportant information from the input image. Finally, the texture of the salient objects is uniformly lit and smoothed out.

We compared the classification accuracy of our proposed method together with the nearest neighbour search against vanilla classification models when the same training data are being used. This way, we can quantitatively estimate the reliability of our proposed method against commonly used models. To this end, we trained baseline classification models (ResNet-50 [82], VGG-11 [81] and MobileNet V2 [83]) as predefined in torchvision on SVIRO-Illumination. For each epoch, we randomly selected one $x_{k}^{j}\in X$ for each piece of scenery $x_{k}$. The classification models were either trained for 1000 epochs or we performed early stopping with a 80:20 split on the training data. We further fine-tuned pretrained models for 1000 epochs. The triplet based autoencoder model was trained exactly as before. During inference, we took the label of the nearest training sample as the classification prediction. Each setup was repeated five times with five different (but the same ones across all setups) seeds. Moreover, the experiments were repeated for all three vehicle interiors. The mean classification accuracy over all five runs, together with the variance, is reported in Table 2. Our proposed method significantly outperforms vanilla classification models trained from scratch and the models’ performances undergo a much smaller variance. Moreover, our proposed method outperforms fine-tuned pretrained classification models, despite the advantage of the pretraining of these models. Additionally, we trained the encoder–decoder models using the vanilla reconstruction error between input and reconstruction, but using the nearest neighbour search as a prediction. Again, including our proposed reconstruction loss improves the models’ performance significantly.

## 7. Synthetic to Real

A small gap between a synthetic and real distribution can potentially be closed by a dedicated data augmentation approach to avoid overfitting to synthetic artefacts. Nevertheless, an abstraction from toy to real images cannot be achieved by means of simple data transformations or model regularizations (e.g., denoising autoencoder). To this end, we proposed using a pretrained feature extractor as presented in Section 3.5 and as defined by Equation (Equation 6). We used the VGG-11 model pretrained on Imagenet as the extractor if not stated otherwise.

### 7.1. SVIRO to TICaM

We trained a vanilla extractor autoencoder with a 64-dimensional latent space on images from the Tesla vehicle from SVIRO and the Kodiaq vehicle from SVIRO-Illumination, respectively, and evaluated the model on the real TICaM images. The transfer from SVIRO to TICaM was further complicated by new unseen attributes, e.g., steering wheel. Examples of the resulting model’s reconstructions are plotted in Figure 10b. In both cases, only blurry human models are reconstructed, which is similar to a mode collapse. We concluded that more robust features are needed.

Hence, we adopted the PIRL because we hypothesized that the same approach could lead to a better generalization to real vehicle interiors. We applied this strategy to variations of the same scene under different illumination conditions, but realized that the learned invariances are not suitable for the transfer between the synthetic and the real. An example is provided in Figure 10d, where we trained on the Kodiaq images from SVIRO-Illumination. Not using the PIRL loss on this dataset performs even worse, i.e., see Figure 10c.

We concluded that, for learning more general features by applying the PIRL, we needed input–target pairs where both images are of the same scene, but differ in the properties we want to become invariant to the dominant background. To this end, we created SVIRO-NoCar—instead of a vehicle, the background was replaced randomly. During training, we randomly select two images per scene and use one as input and the other as the target, i.e., as defined in Equation (Equation 2). When applied to real images, see Figure 10f, the model better preserves the semantics and starts to reconstruct child seats and not people only. We also trained a model without the PIRL to show that the success is not due to the design choice of the dataset: in Figure 10e the model performs worse.

We extended this idea further with our II-PIRL loss formulation: instead of taking the same scene with a different background as the target image, we randomly selected a different scene of the same class, e.g., if a person is sitting at the left seat position, we would take another image with a person on the left seat, potentially a different person with a different pose. This approach is formulated in Equation (Equation 3). While this leads to a blurrier object reconstruction, which is expected because the autoencoder learns an average class representation, the classes are preserved more robustly and the reconstructions look better; see Figure 10g.

The final improvement is based on the assumption that structure in the latent space should help the model performance. Class labels are included by formulating a triplet loss regularization to the latent space representation as defined by Equation (Equation 4). As the results of Figure 10j show, these final improvements, together with the previous changes, yield the semantically most correct reconstructions. The results shows that due to the triplet loss, the nearest neighbour (k) of (j) makes sense and yields a clearer reconstruction.

We investigated whether the qualitative improvements also transfer to a quantitative improvement. We combined E-TAE, I-E-TAE and II-E-TAE, respectively, with a k-nearest neighbour classifier in the latent space, and used our new dataset for training. We retrieve the latent space vectors for all flipped training images as well and used only a single image per scene (i.e., not all 10 variations). We chose $k\;=\;\sqrt{N}\;=\;115$, where *N* is the size of the training data together with its flipped version [84]. We froze the same layers of the pretrained models for fine-tuning the later layers in case of classification models or to train our autoencoder using it as an extractor. We evaluated the model performance after each epoch on the real TICaM images (normal and flipped images of the training and test splits) for both the autoencoder and the corresponding classification model. This provides a measure on the best possible result for each method, but is of course not a valid approach for model selection. We report in Figure 11 the training results for 10 seeds and summarize the training performance by plotting the mean and standard deviation per epoch per method. Our approach converges more robustly and consistently to a better mean accuracy. For each experiment, we retrieve the best accuracy across all epochs and compute the mean, standard deviation and maximum of these values across all runs: these statistics are reported in Table 3. The model weights corresponding to the epochs selected by the previous heuristics were applied on the SVIRO dataset to verify whether the learned representations are universally applicable to other vehicle interiors. For SVIRO, we used the training images and excluded all images containing empty child or infant seats and treated everyday objects as background. The results show that our II-E-TAE significantly outperforms the classification models across three pretrained models and across all datasets.

The triplet loss without the PIRL is not sufficient. An additional ablation study shows that most of the contribution to the success of our introduced model variations stems from the II-PIRL variation. To this end, we trained several types of classifiers in the latent space of different autoencoder model variations and report the results in Table 4. The II-PIRL variation largely improves the classification accuracy compared to the I variation. Moreover, the performance is better compared to the triplet loss variation which uses the labels explicitly in the latent space, compared to the implicit use by the II-PIRL.

### 7.2. MNIST to Real Digits

Finally, we show that improvements reported in the previous section are not limited to the application in the vehicle interior. To this end, we trained models using the same design choices on MNIST [76] and evaluated the generalization onto real digits [85]. Models were trained for 20 epochs using a latent space dimension of 64 and MSE reconstruction loss. As the results in Figure 12 and Table 5 show, similar improvements by the different design choices highlighted on SVIRO can be observed.

## 8. Uncertainty Estimation and Out-of-Distribution Detection

We evaluate our method on two scenarios. First, we want to assess the predictive uncertainty where the model should provide a high uncertainty in case it wrongly classifies a test sample. This is made more difficult in the case of the vehicle interior: unseen new objects should be classified as empty seats, i.e., the model should only identify known classes and neglect everything else. Our results will show that this is a challenging task. Second, the model should differentiate between in- and out-of-distribution (OOD) samples. In the case, for example, of training on MNIST and evaluating on Fashion-MNIST, the model cannot perform a correct prediction and it should detect the OOD as such. This is also the case when images from a new vehicle interior are provided as input to the model. Again, our results will underline the challenge of this task.

In the following section, we use several commonly used computer vision datasets for training. Consequently, we also use the corresponding test data as in-distribution sets $D_{in}$: MNIST, Fashion-MNIST and GTSRB (which we reduce to use 10 classes only). For out-of-distribution $D_{out}$, we use a subset of all $D_{in}$ that do not come from the training distribution and the test datasets from Omniglot [86], SVHN [87], CIFAR10, LSUN [88] (for which we use the train split) and Places365 [89]. We use approximately the same number of samples from $D_{in}$ and $D_{out}$ by sampling each class uniformly. In addition to these commonly used datasets, we also use the training split with adults, child and infants seats from SVIRO-Uncertainty.

We compare our method against autoencoders (MC-AE) and triplet autoencoders (MC-TAE) using the MC dropout approach to highlight that the success is not solely due to the autoencoder training approach. Further, we compare against MC dropout and an ensemble of models using the same architecture as the autoencoder encoder part, but with an additional classification head. All models and methods were trained for 1000 epochs. For training on MNIST and Fashion-MNIST we used a latent space of 16, while for all others we used a latent space of 64. We used 250 samples per class for training and treat all datasets (even RGB ones) as grayscale images. All images were centre cropped and resized to 64 by 64 pixels. We used a dropout rate of 0.33 for all methods and experiments. For MC-AE, MC-TAE, MC-II-AE and MC dropout, we used 20 runs and we used an ensemble of 10 models to assess the uncertainty and OOD estimation. We fixed the seeds for all experiments and we repeated each training method for 10 runs for MC-AE, MC-TAE, MC-II-AE and MC Dropout and for 100 runs to get the ensembles of models, such that we can report mean and standard deviation for the quantitative assessments and comparisons of all methods.

### 8.1. Evaluation Metrics

According to standard evaluation criteria adopted in related works, we evaluate our models using the Area Under the Receiver Operating Characteristic curve (AUROC). While Area Under the Precision-Recall curve (AUPR) and the false positive rate at N% true positive rate (FPR*N*) can also be used, the results were similar. For OOD evaluation, we use approximately 50% of the samples from the test set $D_{in}$ and 50% from the test set from $D_{out}$. For further metric details, we refer to [90,91,92,93].

### 8.2. Results

The results in Table 6 show that the autoencoder model using the II-PIRL always performs better than the vanilla MC-AE or when a triplet loss is used. Notwithstanding this achievement, it can also be observed that the MC-II-AE usually significantly outperforms MC dropout and an ensemble of models. The performance of all methods drops for several SVIRO-Uncertainty splits, which highlights the difficulty of the newly introduced dataset. While the ensemble of models outperforms our method only thrice, it is worth noting that our approach only uses a single model, while the ensemble uses 10 models.

We also compare the reconstructions of the different methods on OOD images when trained on GTSRB in Figure 13. It can be observed that the reconstructions by MC-II-AE are the most clear and diverse in case of OOD input images.

Finally, we also kept track of all entropies for all $D_{in}$ and $D_{out}$. We computed the histograms of the entropies for each dataset and each method and report results in Figure 14 when trained on GTSRB. The results show that the entropy distribution between $D_{in}$ and several $D_{out}$ are best separated when the II-PIRL is used. The distributions of the different $D_{out}$ are more similar then for the other methods. We computed the sum of the Wasserstein distances between $D_{in}$ and all $D_{out}$ (TD, larger is better, because we want them to be different) separately and the sum of the distances between $D_{out}$ CIFAR10 and all other $D_{out}$ (OD, smaller is better, because we want them to be similar). We then computed the mean and standard deviation across the 10 runs and report the results in Table 7. The results for MC-AE, MC-TAE, MC-II-AE, MC dropout and an ensemble of 10 models show that our method best separates uncertainty between $D_{in}$ and $D_{out}$. Further, all $D_{out}$ are most similar between each other for our method.

## 9. Discussion and Future Work

The II variation of the PIRL loss implicitly assumes that the classes are unimodal, i.e., objects of the same class should be mapped onto a similar point in the latent space. This characteristic can either improve generalization or have a detrimental effect on the performance depending on the task to be solved. Under its current form there is no guarantee that, for example, facial landmarks or poses would be preserved. Nevertheless, we believe that extensions of our proposed loss, for example based on incorporating constraints (e.g., preservation of poses and landmarks) could be an interesting direction for future work.

Another interesting direction for future research would be the investigation of resulting properties when the PIRL is used during training against other sampling strategies. It would be interesting to see whether PIRL implicitly forces the learned training data manifold in the latent space to follow some beneficial properties, e.g., Euclidean space or disentanglement.

### 9.1. Normalization

The I-PIRL works well for image normalization on the training data, which can be sufficient for some applications, e.g., when a fixed dataset is available on which some post-processing needs to be performed exclusively. Since the generalization to test images can be achieved by a nearest neighbour search, the latter will only be useful for a subset of machine learning tasks. Our method preserves the classes for a given problem formulation, which will be fine for classification and object detection. Our method can preserve head poses to some extent (e.g., Figure 5) when it is dominantly present in the images.

In practise, it will be challenging to record identical sceneries under different lightning conditions. However, as the Extended Yale Face Database B [24] and Webcam Clip Art [28] dataset have shown, it is also feasible. Since we have highlighted the benefit of the acquisition of said datasets, the investment of recording under similar conditions in practise can be worthwhile for some applications. We believe that future work will develop possibilities to facilitate the data acquisition process.

### 9.2. Synthetic to Real

While neither the PIRL nor the extractor provide the best results alone, the combination of both is a beneficial design choice. It leads to an advantageous transferability and the training more robustly achieves better accuracies. It can be observed that our model is not perfect and sometimes struggles, e.g., in case an object (e.g., backpack) is located on the seat and for more complex human poses (e.g., people turning over). It is clear that the reconstruction is far from perfect, and hence it might not be suitable for some applications. However, we believe that these problems are related to the training data: a more versatile synthetic dataset could improve the model performance on more challenging real images.

### 9.3. Uncertainty

As our results show, the II-PIRL can improve OOD and uncertainty estimation compared to other autoencoder variations. Additionally, our approach can also outperform commonly used uncertainty estimation approaches for deep learning models. While it also performs well on SVIRO-Uncertainty, the performance is far from the results on MNIST, Fashion-MNIST and GTSRB. Potentially, the reason might be the comparatively small dataset size compared to the other datasets. Nevertheless, II-PIRL can easily be adapted to commonly used datasets and combined with our approaches to further improve the performance.

## 10. Conclusions

We investigated several questions stemming from industry, but with repercussions for many machine learning applications. We broke down the problems into fundamental questions and research directions: invariances with respect to illumination and background, the transfer from synthetic to real sceneries, uncertainty estimation and, in general, the extraction of reliable features for downstream computer vision classification tasks. The results of an exhaustive ablation study on several computer vision tasks and datasets showed that autoencoders using both of our proposed partially impossible reconstructions losses have a beneficial effect on the model performance. Our novel loss variations also induce interesting reconstruction phenomena: most of the unimportant information is removed or smoothed out, which can potentially be exploited by downstream tasks. We discussed the benefits and downsides of both loss variations and showed that they produce good latent space representations on par with the triplet loss.

The first, weaker variation of the PIRL exploits the availability of certain dataset characteristics: if, for each piece of scenery, several variations regarding the invariances to be learned are available, e.g., same scenery under different illumination conditions, then the unwanted features can be removed reliably without affecting human poses. The second, stronger variation only uses label information: for each input image, a target image of the same class is sampled randomly. This induces robust feature extraction, allowing for higher classification accuracies. The latter design choice can also be used to improve the transfer from synthetic to real images and to provide more reliable uncertainty estimations. However, human poses cannot be preserved under its current form. In order to learn more general features, we proposed the use of pretrained convolutional neural networks to extract features from the input images, which were then used as input by the autoencoder model. This improved synthetic to real generalization, particularly if combined with the second variation of the partially impossible reconstruction loss and the triplet loss.

Our proposed sampling strategies can easily be combined with any existing autoencoder model, reconstruction loss and potential latent space regularization. Overall, a partially impossible reconstruction loss seems beneficial to learn more general and reliable features on a wide range of tasks.

## Figures and Tables

**Figure 1 sensors-22-04862-f001:**
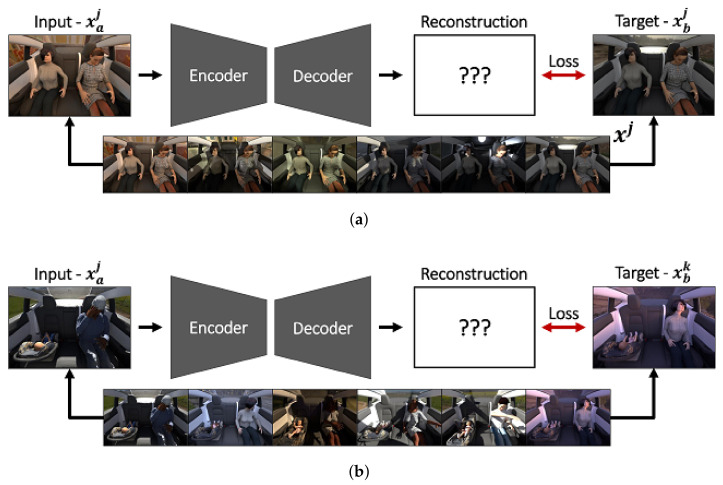
Illustration of an autoencoder model (encoder and decoder) using the two different partially impossible reconstruction losses together with sampling examples. (**a**) Variation I: input $X_{a}^{j}$ and target $X_{b}^{j}$ are images of the same scenery $X^{j}$, but under different illumination conditions. (**b**) Variation II: for the input $X_{a}^{j}$ a new scenery $X_{b}^{k}$ of the same class label is selected as target.

**Figure 2 sensors-22-04862-f002:**
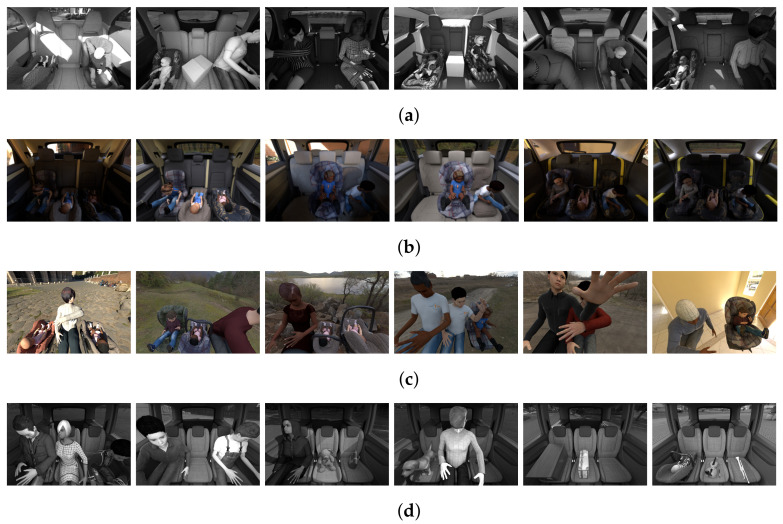
Example images for the different SVIRO datasets variations. Each seat position (left, middle and right) can be empty, or an adult, a child seat, an infant seat or an everyday object can be placed on it. (**a**) SVIRO: Base dataset with sceneries for ten different vehicle interiors. (**b**) SVIRO-Illumination: Scenery is rendered under several illuminations conditions. (**c**) SVIRO-NoCar: Scenery is rendered with different background and without a vehicle. (**d**) SVIRO-Uncertainty: Fine-grained splits with unknown everyday objects for uncertainty estimation.

**Figure 3 sensors-22-04862-f003:**
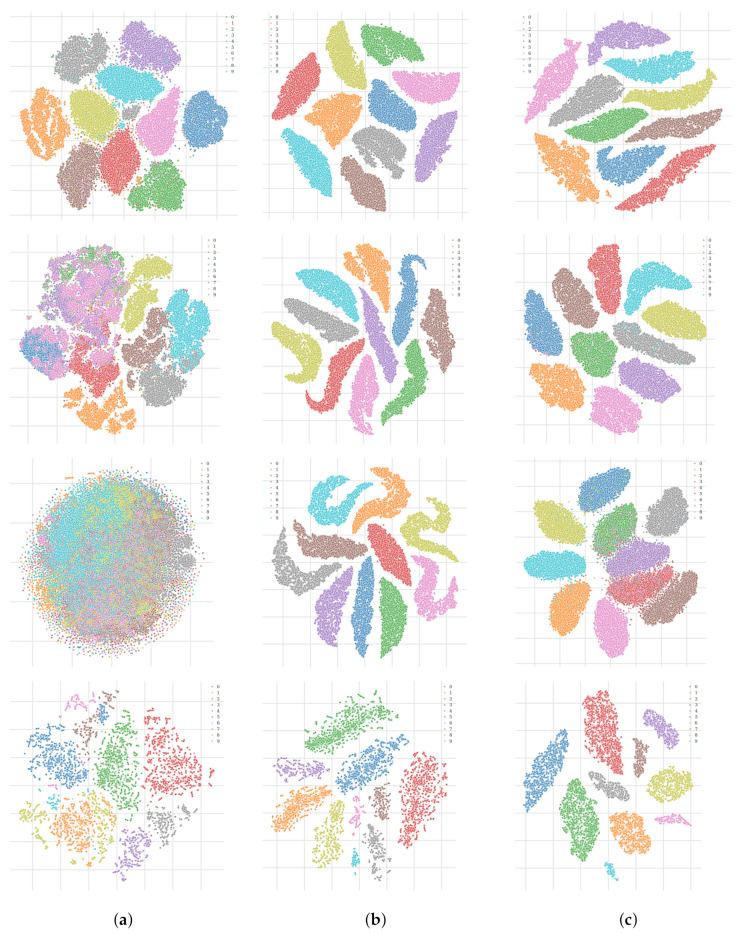
Comparison of training latent space representations (t-SNE projection) by different autoencoder models for different datasets: MNIST (first block), Fashion-MNIST (second block), CIFAR10 (third block) and GTSRB (fourth block). We either used the triplet loss (TAE), the second variation of the PIRL (II-AE) or just the vanilla reconstruction loss (AE). Different colours represent different classes. (**a**) AE. (**b**) TAE. (**c**) II-AE (Ours).

**Figure 4 sensors-22-04862-f004:**
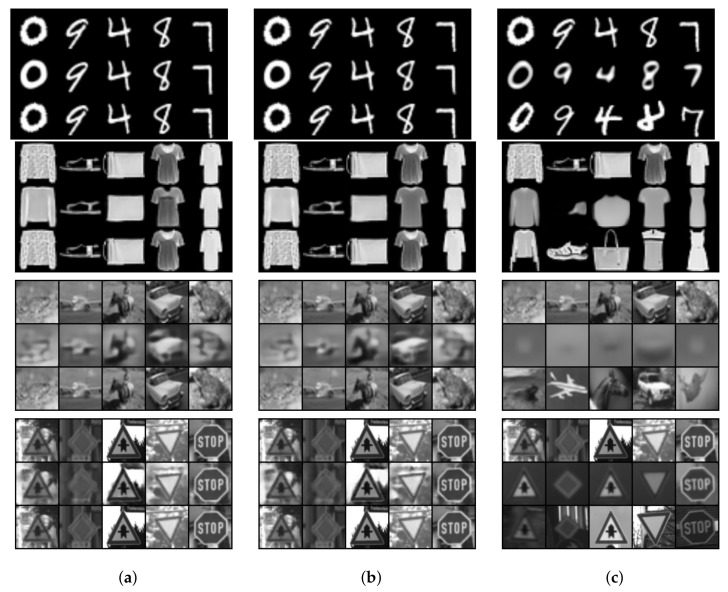
Comparison of the input (first row), reconstruction (second row) and target (third row) of the training data after the last epoch. The results are for different autoencoder models for different datasets: MNIST (first block), Fashion-MNIST (second block), CIFAR10 (third block) and GTSRB (fourth block). We either used the triplet loss (TAE), the second variation of the PIRL (II-AE) or just the vanilla reconstruction loss (AE). (**a**) AE. (**b**) TAE. (**c**) II-AE (Ours).

**Figure 5 sensors-22-04862-f005:**
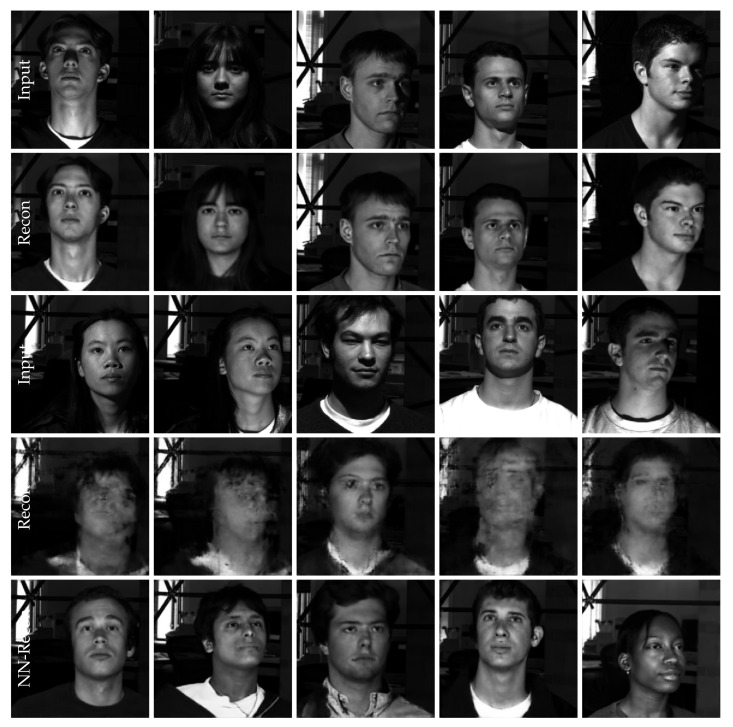
Extended Yale Face Database B. Illumination is removed from the training samples (first row) to form the reconstruction (second row). The test samples (third row) cannot always be reconstructed reliably (fourth row). Reconstructing the nearest neighbour (fifth row) preserves the head pose and position and removes the illumination such that potentially a classification or a landmark detection could be performed on the latter instead of the input image.

**Figure 6 sensors-22-04862-f006:**
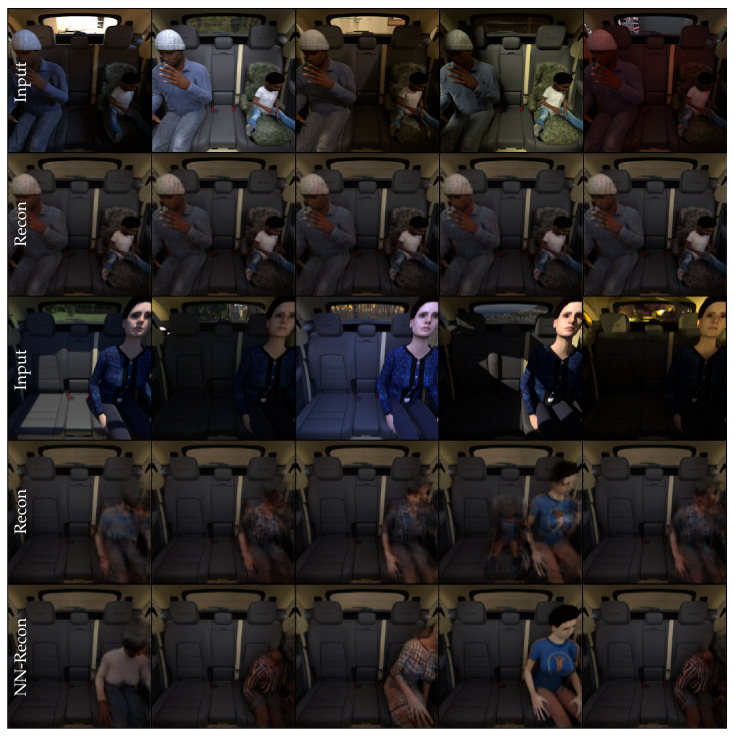
The encoder–decoder model transforms the input images of the same scenery (first row) into a cleaned version (second row) by removing all illumination and environment information (see the background through the window). The test image (third row) cannot be reconstructed perfectly (fourth row). Choosing the nearest neighbour in the latent space and reconstructing the latter leads to a class-preserving reconstruction (fifth row).

**Figure 7 sensors-22-04862-f007:**
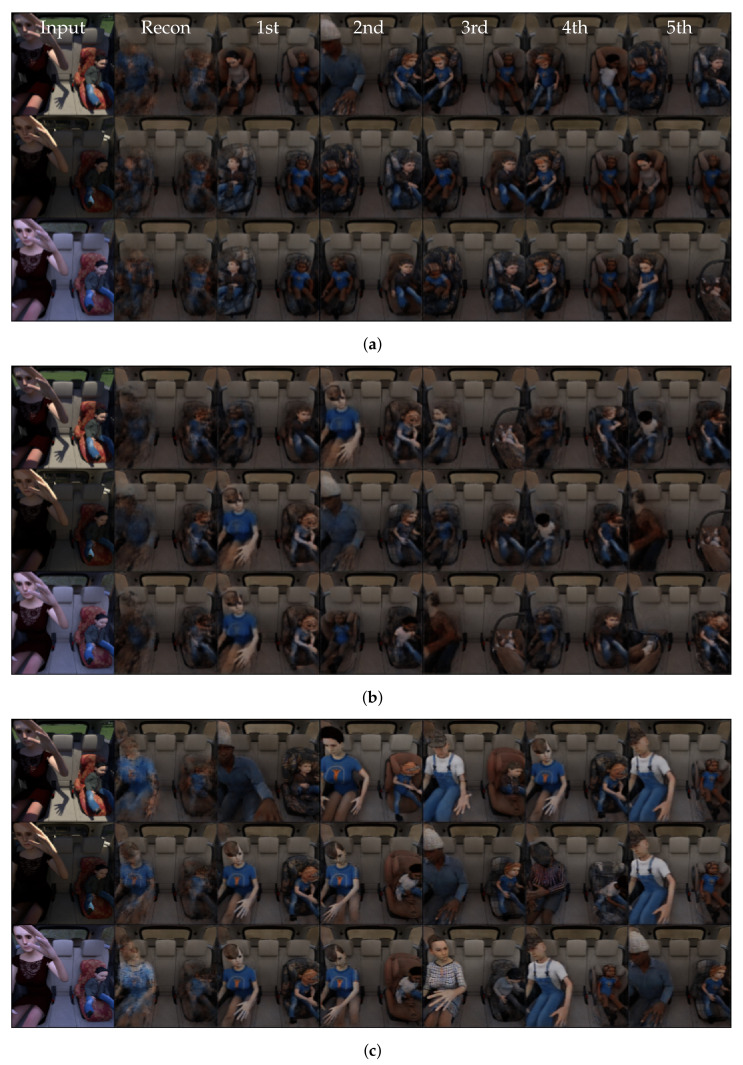
Comparison of the reconstruction of the 5 nearest neighbours (columns 3 to 7) for different encoder–decoder latent spaces (**a**–**c**). The reconstruction (second column) of the test sample (first column) is also reported. We used the I-PIRL and either used the triplet loss (TAE), the variational autoencoder (VAE) or just the vanilla reconstruction loss (AE). The triplet regularization is the most reliable and consistent one across all 5 neighbours. Notice the class changes across neighbours for the AE and VAE models. (**a**) I-AE. (**b**) I-VAE. (**c**) I-TAE.

**Figure 8 sensors-22-04862-f008:**
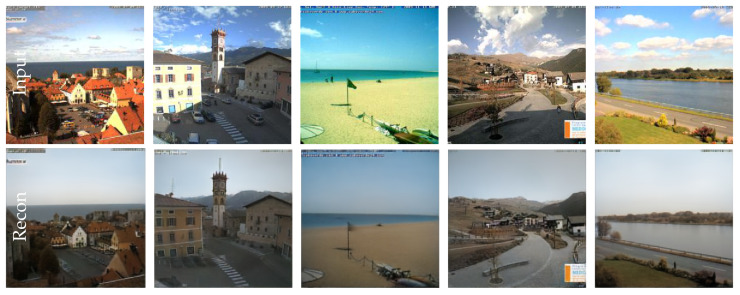
Examples for the Webcam Clip Art dataset. The encoder–decoder model removes the environmental features from the images (**first row**) to form the output images (**second row**). Vehicles and people are removed from the scenery and skies, rivers and beaches are smoothed out.

**Figure 9 sensors-22-04862-f009:**
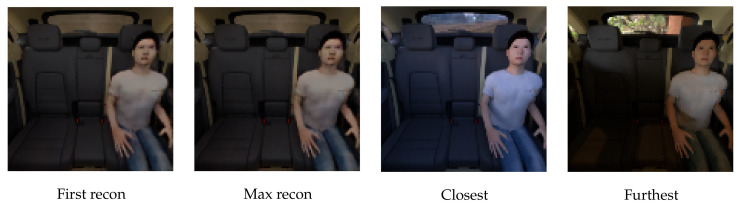
Reconstruction of the first scenery (first recon) is compared against the furthest reconstruction of all sceneries (max recon). First recon is also used to determine the closest and furthest scenery. The model does not learn to focus the reconstruction to a training sample.

**Figure 10 sensors-22-04862-f010:**
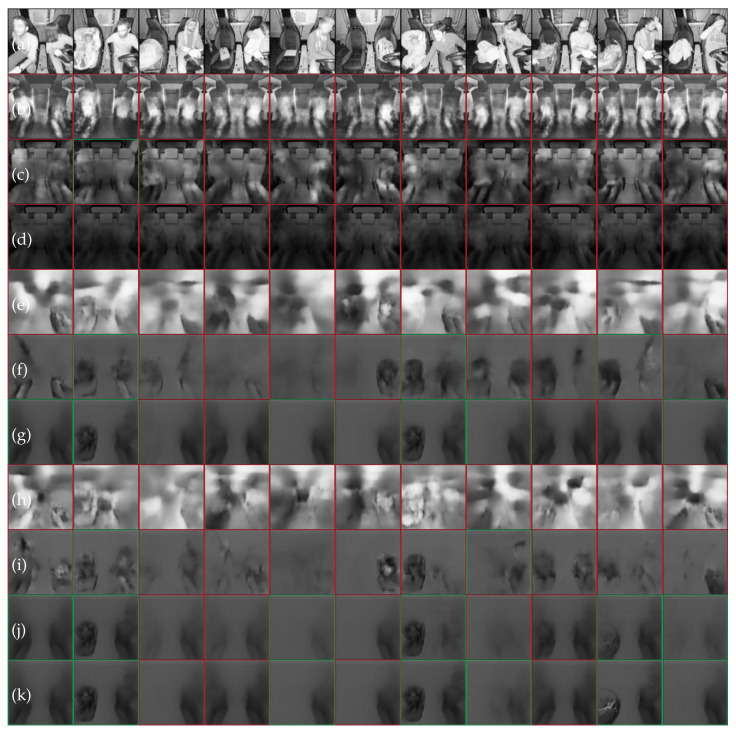
Reconstruction results of unseen real data (**a**) from the TICaM dataset: (**b**) E-AE Trained on Tesla SVIRO, (**c**) E-AE Trained on Kodiaq SVIRO-Illumination, (**d**) I-E-AE Trained on Kodiaq SVIRO-Illumination, (**e**) E-AE, (**f**) I-E-AE, (**g**) II-E-AE, (**h**) E-TAE, (**i**) I-E-TAE, (**j**) II-E-TAE and (**k**) Nearest neighbour of (**j**). Examples (**e**–**k**) are all trained on our new dataset. A red (wrong) or green (correct) box highlights whether the semantics are preserved by the reconstruction.

**Figure 11 sensors-22-04862-f011:**
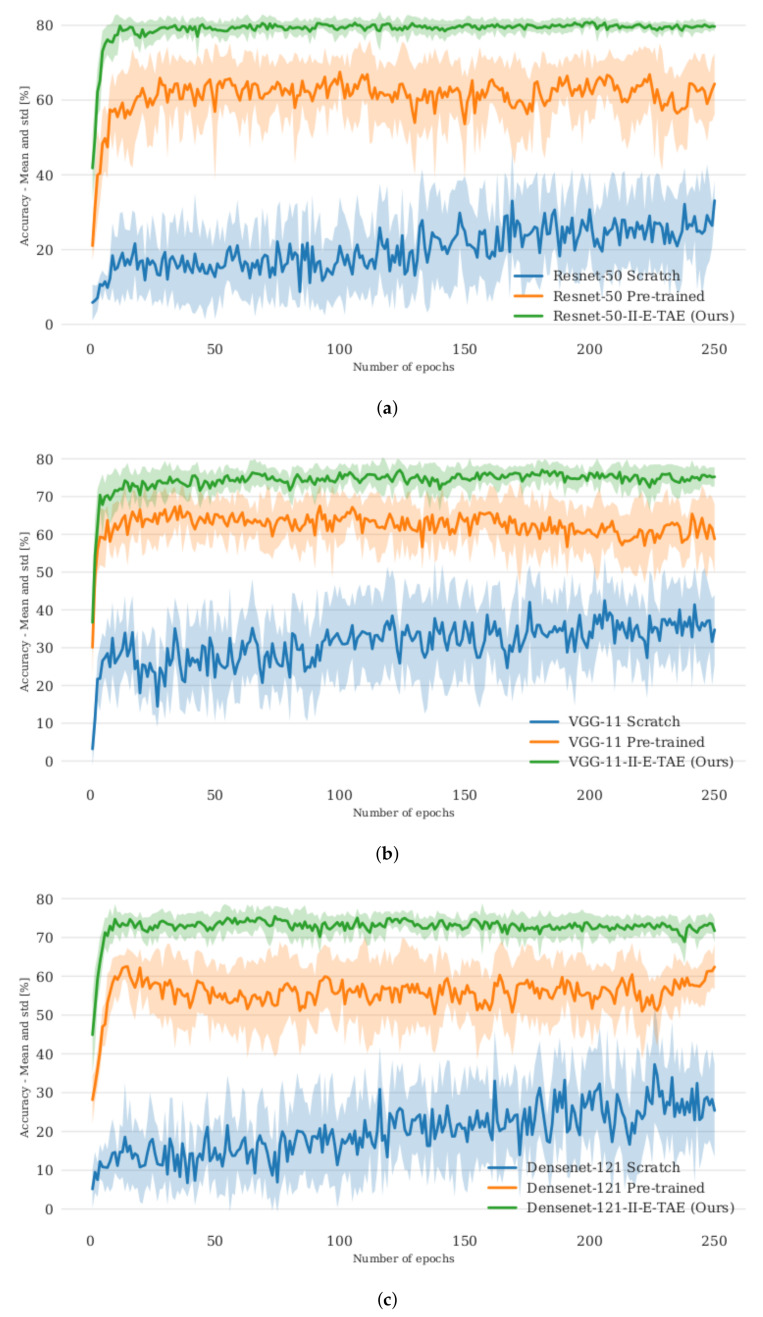
Comparison of the training performance distribution for each epoch over 250 epochs. II-E-TAE (extractor triplet autoencoder using the II-PIRL) is compared against training the extractor from scratch or fine-tuning the layers after the features used by the extractor in our approach. (**a**) Resnet-50. (**b**) VGG-11. (**c**) Densenet-121.

**Figure 12 sensors-22-04862-f012:**
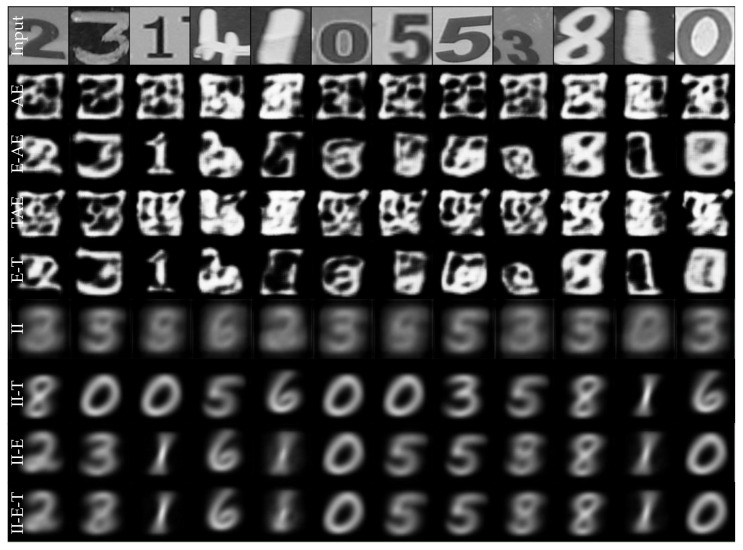
Reconstruction of real digits when trained on MNIST. We either used the triplet loss (T or TAE), the second variation of the PIRL (II) and/or the extractor (E) or just the vanilla reconstruction loss (AE). The II-PIRL provides the best class preserving reconstructions.

**Figure 13 sensors-22-04862-f013:**
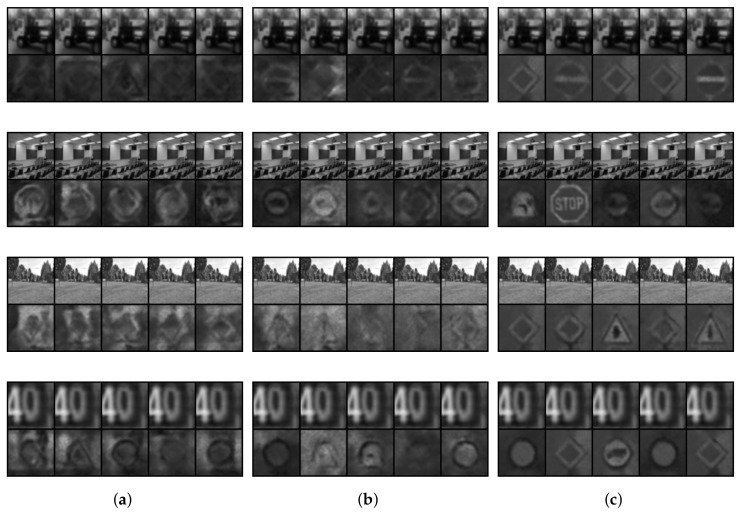
Comparison of OOD images (first row) with the corresponding reconstructions (second row) for several inferences when dropout is enabled. The results are for different autoencoder models for different datasets: CIFAR10 (first block), LSUN (second block), Places365 (third block) and SVHN (fourth block). We either used the triplet loss (TAE), the second variation of the PIRL (II-AE) or just the vanilla reconstruction loss (AE). (**a**) MC-AE. (**b**) MC-TAE. (**c**) MC-II-AE (Ours).

**Figure 14 sensors-22-04862-f014:**
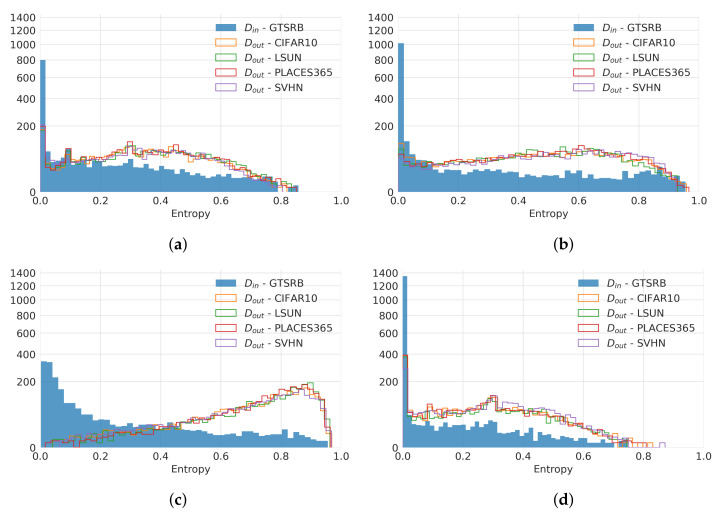
Comparison of entropy histograms between $D_{in}$ (GTSRB, filled bars with blue) and several $D_{out}$ (not filled bars and coloured according to dataset used) for different methods. We either used the triplet loss (TAE), the second variation of the PIRL (II-AE) or just the vanilla reconstruction loss (AE) and compare them against MC Dropout and an ensemble of models. MCA-II-AE provides the best separation between $D_{in}$ (in-distribution) and $D_{out}$ (out-of-distribution). Moreover, the different $D_{out}$ have a more similar distribution compared to the other models, which we also evaluate quantitatively. Notice the non-linear scale on the y-axis (number count per bin) to ease visualization for smaller values. (**a**) MC-AE. (**b**) MC-TAE. (**c**) MC-II-AE (Ours). (**d**) MC Dropout. (**e**) Ensemble.

**Table 1 sensors-22-04862-t001:** Mean and standard deviation test accuracy over 10 runs by a linear SVM classifier trained in the latent space of different autoencoders after the latter finished training. We either used the triplet loss (TAE), the second variation of the PIRL (II-AE) or just the vanilla reconstruction loss (AE).

Dataset	II-AE (Ours)	TAE	AE
MNIST	99.3 ± 0.1	99.1 ± 0.1	93.2 ± 0.5
Fashion	91.9 ± 0.2	90.7 ± 0.3	78.9 ± 0.3
CIFAR10	65.9 ± 0.6	60.4 ± 1.4	18.6 ± 2.0
GTSRB	98.9 ± 0.4	98.8 ± 0.3	95.7 ± 0.5

**Table 2 sensors-22-04862-t002:** Mean accuracy and variance over 5 repeated training runs on each of the three vehicle interiors. F—fine-tuned pretrained model, ES—early stopping with 80:20 split, NS—no early stopping and V—vanilla reconstruction loss. We either used the triplet loss (TAE), the second variation of the PIRL (II-AE) or just the vanilla reconstruction loss (AE). PIRL improves the vanilla version and, with the nearest neighbour search, outperforms all other models.

	Vehicle		
**Model**	**Cayenne**	**Kodiaq**	**Kona**
MobileNet-ES	62.9 ± 3.1	71.8 ± 4.3	73.0 ± 0.8
VGG11-ES	64.4 ± 35.0	74.0 ± 19.0	75.5 ± 5.7
ResNet50-ES	72.3 ± 3.7	77.9 ± 35.0	76.6 ± 9.9
MobileNet-NS	72.7 ± 3.8	77.0 ± 4.1	77.4 ± 2.2
VGG11-NS	74.1 ± 5.8	71.2 ± 14.0	78.4 ± 2.6
ResNet50-NS	76.2 ± 18.0	83.1 ± 1.1	82.0 ± 3.2
MobileNet-F	85.8 ± 2.0	90.6 ± 1.2	88.6 ± 0.6
VGG11-F	90.5 ± 2.0	90.3 ± 1.2	89.2 ± 0.9
ResNet50-F	87.9 ± 2.0	89.7 ± 6.1	88.5 ± 1.0
AE-V	74.1 ± 0.7	80.1 ± 1.8	73.3 ± 0.9
VAE-V	73.4 ± 1.3	79.5 ± 0.6	73.0 ± 0.9
TAE-V	90.8 ± 0.3	91.7 ± 0.2	89.9 ± 0.6
I-AE (Ours)	86.8 ± 0.3	86.7 ± 1.5	86.7 ± 0.9
I-VAE (Ours)	81.4 ± 0.5	86.6 ± 0.9	85.9 ± 0.8
I-TAE (Ours)	92.4 ± 1.5	93.5 ± 0.9	93.0 ± 0.3

**Table 3 sensors-22-04862-t003:** For each experiment, the best performance on real vehicle interior images (TICaM) across all epochs is taken and then the mean and maximum of those values across all 10 runs is reported. The model weights achieving maximum performance per run are evaluated on SVIRO, where they perform better as well. We used the triplet loss (TAE), the first (I) or the second variation (II) of the PIRL and the extractor module (E).

Dataset	TICaM		SVIRO	
Model	Mean	Max	Mean	Max
VGG				
Scratch	58.5 ± 4.0	64.6	65.6 ± 5.4	72.7
Pretrained	75.5 ± 1.5	78.0	78.7 ± 2.9	84.0
E-TAE	76.7±2.3	81.5	78.6 ± 2.6	82.3
I-E-TAE	79.7 ± 2.1	82.2	80.9 ± 4.0	85.6
II-E-TAE	81.0 ± 0.6	82.0	79.1 ± 3.9	84.8
Resnet				
Scratch	53.3 ± 3.5	60.4	56.4 ± 2.6	59.3
Pretrained	78.1 ± 1.7	80.4	83.5 ± 2.7	88.1
E-TAE	83.8 ± 1.3	86.0	85.8 ± 2.4	89.1
I-E-TAE	83.5 ± 1.3	85.6	89.2 ± 1.0	90.3
II-E-TAE	83.7 ± 0.5	84.5	93.0 ± 0.8	94.1
Densenet				
Scratch	56.3 ± 5.5	62.1	68.8 ± 2.4	74.9
Pretrained	72.2 ± 4.2	77.4	85.0 ± 2.3	88.0
E-TAE	78.5 ± 2.4	81.8	86.7 ± 1.3	88.2
I-E-TAE	77.2 ± 1.7	79.3	90.4 ± 1.3	92.1
II-E-TAE	79.3 ± 1.3	81.5	89.9 ± 1.8	92.3

**Table 4 sensors-22-04862-t004:** For each of the 10 experimental runs per method after 250 epochs and using the VGG-11 extractor, we trained different classifiers in the latent space: k-nearest neighbour (KNN), random forest (RForest) and support vector machine with a linear kernel (SVM). We either used the triplet loss (TAE), the first (I) or the second variation (II) of the PIRL and the extractor module (E). Most of the contribution to the synthetic to real generalization on TICaM is due to the II variation of the PIRL cost function.

Variant	KNN	RForest	SVM
E-AE	17.1 ± 6.7	24.2 ± 4.1	40.6 ± 8.5
I-E-AE	18.2 ± 7.3	42.4 ± 6.5	50.1 ± 3.7
II-E-AE	73.2 ± 3.9	68.8 ± 5.7	66.9 ± 6.7
E-TAE	69.2 ± 3.4	66.4 ± 4.0	68.7 ± 2.2

**Table 5 sensors-22-04862-t005:** Different model variations trained on MNIST. The classifiers were trained on the training data latent space and evaluated on real digits: k-nearest neighbour (KNN), random forest (RForest) and support vector machine with a linear kernel (SVM). We used the triplet loss (TAE), the first (I) or the second variation (II) of the PIRL and the extractor module (E).

Model	KNN	RForest	SVM
AE	15.7	12.5	11.6
TAE	11.1	11.6	8.4
II-AE	27.8	20.2	23.6
II-TAE	21.8	17.9	23.9
E-AE	27.3	23.1	26.5
E-TAE	26.1	19.1	23.3
II-E-AE	65.0	61.9	65.6
II-E-TAE	64.1	63.7	63.7

**Table 6 sensors-22-04862-t006:** Comparison of AUROC (in percentage, larger is better) of our method against MC dropout and an ensemble of models (Ensemble) as well as vanilla (AE) and triplet autoencoders (TAE) either using the second variation of the PIRL (II) or not. We repeated the experiments for 10 runs and report the mean and standard deviation. If $D_{in}\;=\;D_{out}$, then we report the result on the test set of $D_{in}$. $D_{in}$ is in-distribution data and $D_{out}$ out-of-distribution data. Best results are highlighted in grey.

$D_{\mathrm{in}}\to D_{\mathrm{out}}$	MC-II-AE (Ours)	MC-TAE	MC-AE	MC Dropout	Ensemble
MNIST → MNIST	93.6 ± 0.4	93.3 ± 1.0	84.9 ± 0.8	90.2 ± 0.8	81.0 ± 1.6
MNIST → CIFAR10	99.1 ± 0.8	97.5 ± 1.1	81.0 ± 5.1	91.8 ± 1.8	91.9 ± 1.8
MNIST → Fashion	97.3 ± 0.7	95.0 ± 1.1	77.2 ± 6.1	88.5 ± 2.4	82.6 ± 2.5
MNIST → Omniglot	99.4 ± 0.4	97.6 ± 0.7	82.6 ± 9.0	93.2 ± 4.0	95.8 ± 2.3
MNIST → SVHN	99.1 ± 1.1	98.1 ± 1.0	81.5 ± 7.1	94.9 ± 1.9	94.2 ± 1.6
Fashion → Fashion	86.2 ± 0.5	85.8 ± 0.8	83.5 ± 0.8	82.1 ± 0.4	79.8 ± 0.8
Fashion → CIFAR10	96.6 ± 1.3	91.7 ± 1.9	91.2 ± 3.2	88.6 ± 1.2	91.0 ± 1.0
Fashion → MNIST	91.5 ± 1.7	87.2 ± 2.3	76.4 ± 6.4	83.2 ± 2.0	88.4 ± 0.8
Fashion → Omniglot	97.7 ± 1.2	89.0 ± 3.0	77.7 ± 8.7	91.7 ± 2.4	96.9 ± 0.9
Fashion → SVHN	95.7 ± 2.5	90.5 ± 2.7	92.1 ± 3.4	90.0 ± 1.1	93.6 ± 1.1
GTSRB → GTSRB	94.6 ± 0.9	93.4 ± 0.8	87.9 ± 1.6	85.7 ± 1.2	83.2 ± 0.9
GTSRB → CIFAR10	92.6 ± 3.3	80.7 ± 1.8	75.4 ± 2.5	79.0 ± 0.8	69.2 ± 1.0
GTSRB → LSUN	93.9 ± 3.5	81.2 ± 1.9	76.6 ± 2.3	80.3 ± 0.6	68.3 ± 0.8
GTSRB → Places365	93.6 ± 3.6	82.0 ± 1.7	76.2 ± 2.1	79.4 ± 0.5	68.7 ± 0.8
GTSRB → SVHN	92.8 ± 3.0	83.6 ± 2.4	76.1 ± 3.7	82.8 ± 1.2	72.7 ± 0.7
SVIRO-U → CIFAR10	87.2 ± 7.2	71.5 ± 21.8	78.5 ± 4.9	70.8 ± 10.6	83.8 ± 2.2
SVIRO-U → GTSRB	74.5 ± 9.4	68.1 ± 18.4	82.5 ± 4.8	76.6 ± 6.2	87.0 ± 1.9
SVIRO-U → LSUN	84.4 ± 8.2	71.0 ± 21.2	77.5 ± 4.4	74.0 ± 8.5	82.7 ± 1.8
SVIRO-U → Places365	85.7 ± 7.8	71.2 ± 21.4	79.4 ± 3.8	75.3 ± 7.4	83.7 ± 1.5
SVIRO-U → SVHN	92.7 ± 4.9	72.4 ± 22.6	79.0 ± 4.8	66.2 ± 13.0	84.4 ± 3.0
SVIRO-U → Adults (A)	97.6 ± 0.8	48.7 ± 48.7	88.4 ± 1.1	91.9 ± 0.8	87.3 ± 0.7
SVIRO-U → Seats (S)	93.3 ± 2.8	46.4 ± 46.4	74.9 ± 2.1	89.7 ± 4.2	89.0 ± 2.8
SVIRO-U → Objects (O)	75.5 ± 4.2	39.5 ± 39.6	73.4 ± 1.6	73.7 ± 2.5	73.4 ± 4.5
SVIRO-U → A, S	92.4 ± 2.0	45.7 ± 45.7	75.0 ± 1.7	87.8 ± 1.7	79.0 ± 1.7
SVIRO-U → A, O	81.8 ± 2.3	66.2 ± 16.2	76.4 ± 1.0	80.5 ± 1.9	81.8 ± 0.8
SVIRO-U → A, S, O	77.1 ± 2.6	37.9 ± 37.9	69.6 ± 1.2	75.3 ± 1.9	77.8 ± 1.3
SVIRO-U → Tesla (OOD)	79.0 ± 11.8	65.3 ± 16.5	72.7 ± 4.8	81.3 ± 3.8	80.3 ± 2.2

**Table 7 sensors-22-04862-t007:** We calculated the sum of the Wasserstein distances between $D_{in}$ (in-distribution) and all $D_{out}$ (TD, larger is better) separately and the sum of the distances between $D_{out}$ (out-of-distribution) CIFAR10 and all other $D_{out}$ (OD, smaller is better) over 10 runs. We report the mean and standard deviation and compare our method (MC-II-AE) against MC dropout and an ensemble of models (Ensemble) as well as vanilla (AE) and triplet autoencoders (TAE).

		MC-II-AE (Ours)	MC-TAE	MC-AE	MC Dropout	Ensemble
OD	↓	0.082 ± 0.026	0.086 ± 0.025	0.041 ± 0.014	0.047 ± 0.014	0.030 ± 0.004
TD	↑	1.916 ± 0.332	1.261 ± 0.124	0.782 ± 0.112	0.749 ± 0.014	0.436 ± 0.020

## Data Availability

SVIRO and all its extensions can be found on and downloaded from our website—https://sviro.kl.dfki.de/, accessed on 1 May 2022. The code implementation to reproduce the results presented in this work can be found in the Appendix A.

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
