# Peer review of "Autoencoder and Partially Impossible Reconstruction Losses"

_sensors, 2022, doi:10.3390/s22134862_

Round 1

Reviewer 1 Report

  • The abstract explains the meaning of the paper and includes the background parts, but it lacks a clear emphasis on the results obtained. The abstract is well written, but it would be good to have more emphasis on the specific results obtained in this manuscript.
  • The introduction provides a good general background to the topic and gives the reader an idea of the wide range of possible applications of this technology. Reading the manuscript as a whole, 30 pages of text is still a bit too much for a manuscript. For this very reason, certain chapters of the manuscript do not come to the fore. For this reason, it is suggested that authors try to shorten individual chapters where possible.
  • The methods used in this paper are appropriate to the aim of the study. Results are mostly clearly explained and presented in an appropriate format.
    • During the presentation of the results, many shortcomings were noted. Namely, many titles do not contain enough information to easily follow the results (e.g., Figure 1; Figure 2; Figure 3; Figure 4; Figure 5; etc.). In addition, for many images, the authors did not list the titles on the x- and y-axes, which certainly makes it difficult to understand the results shown (e.g., Figure 3, Figure 14), while for some images the font is so small that it is difficult to read the results (Figure 11).
    • To understand the table shown, it is necessary to place an explanation of all the parameters shown in it at the foot of the table. When viewing the results, the reader should independently follow the results shown, without paying attention to abbreviations in the rest of the text.
  • The conclusions in this paper need improvement. It is not clear how your research contributed to knowledge gaps, and there is no information about research limitations for future research.

Reviewer 2 Report

This paper introduced two variations of partially impossible reconstruction losses (PIRL) to train autoencoders for salient feature extraction and neglection of unimportant parts of the images. The proposed methods were applied to and systematically investigated for various computer vision tasks. Overall, this is an interesting paper, and I only have a few comments regarding clarifications.

1. First sentence in Sec 3.3. “The previously introduced PIRL will lead to good results, particularly for normalization, however, it can be challenging to apply it to a lot of commonly recorded datasets.” The authors should explain why the introduced PIRL (I-PIRL) can be challenging for many recorded datasets. Also, the description of motivation for proposing the second variation of PIRL (II-PIRL) should be more precise.

2. In Sec 3.4.: “the explicit use of labels in the latent space induces an L2 metric”. I don’t understand what this sentence means. Please give a clear explanation.

3. The loss function defined above line 194 lacks an equation number. Also, how were the parameters α and β determined? Moreover, it will be interesting to investigate the effect of the proposed loss function when different α and β are used.

4. Line 285: “We treat all datasets (even RGB ones) as grayscale images”. Please describe how you used 3-channel RGB images as 1-channel grayscale images.

5. In Sec 5, the authors trained a linear classifier to quantitatively compare the separability of the learned latent space representation. However, only using a linear classifier is limited, and there might be a case in which the linear classifier produces smaller accuracy, but the nonlinear classifier produces greater accuracy that indicates the latent space representation has better separability. It will be interesting to train an extra nonlinear classifier to compare the latent space representation's separability.

6. Line 321: “Consequently, instead of reconstructing the encoded test sample, it is more beneficial to reconstruct its nearest neighbour.” Please explain why reconstructing the nearest neighbor is more beneficial than reconstructing the encoded test sample itself. Also, is the nearest neighbor with respect to SSIM? The authors should provide the definition of the nearest neighbor they used.

7. Figure 5: Although reconstructing the nearest neighbor can produce clean images, it does not achieve the original purpose of illumination removal because the output image does not represent the same object (e.g., face) being imaged. Please explain why reconstructing the nearest neighbor is beneficial if the original purpose cannot be achieved.
